# Uncertainty Estimation for Computed Tomography with a Linearised Deep Image Prior

**Javier Antorán**[*]                                                   *ja666@cam.ac.uk*
*Department of Engineering, University of Cambridge*

**Riccardo Barbano**[*]                                              *riccardo.barbano@tum.de*
*School of Computation, Information and Technology, Technical Univeristy of Munich*

**Johannes Leuschner**                                             *jleuschn@uni-bremen.de*
*Center for Industrial Mathematics, University of Bremen*

**José Miguel Hernández-Lobato**                                *jmh233@cam.ac.uk*
*Department of Engineering, University of Cambridge*

**Bangti Jin**                                                         *b.jin@cuhk.edu.hk*
*Department of Mathematics, The Chinese University of Hong Kong*

**Reviewed on OpenReview:** *https://openreview.net/forum?id=FWyabz82fH*

## Abstract

Existing deep-learning based tomographic image reconstruction methods do not provide accurate uncertainty estimates of their reconstructions, hindering their real-world deployment. This paper develops a method, termed as linearised deep image prior (DIP), to estimate the uncertainty associated with reconstructions produced by the DIP with total variation (TV) regularisation. We endow the DIP with conjugate Gaussian-linear model type error-bars computed from a local linearisation of the neural network around its optimised parameters. To preserve conjugacy, we approximate the TV regulariser with a Gaussian surrogate. This approach provides pixel-wise uncertainty estimates and a marginal likelihood objective for hyperparameter optimisation. We demonstrate the method on synthetic data and real-measured high-resolution 2D $\mu$CT data, and show that it provides superior calibration of uncertainty estimates relative to previous probabilistic formulations of the DIP. Our code is available at https://github.com/educating-dip/bayes_dip.

## 1 Introduction

Inverse problems in imaging aim to recover an unknown image $x \in \mathbb{R}^{d_x}$ from the noisy measurement $y \in \mathbb{R}^{d_y}$

$$y = \mathrm{A}x + \eta, \qquad (1)$$

where $\mathrm{A} \in \mathbb{R}^{d_y \times d_x}$ is a linear forward map, and $\eta$ i.i.d. Gaussian noise, i.e. $\eta \sim \mathcal{N}(0, \sigma_y^2 \mathrm{I})$. Many tomographic reconstruction problems take this form, e.g. computed tomography (CT). Due to the inherent ill-posedness of the problem, e.g. $d_y \ll d_x$, suitable regularisation / prior is crucial for the successful recovery of $x$ (Tikhonov & Arsenin, 1977; Engl et al., 1996; Ito & Jin, 2014).

Figure 1: X-ray reconstruction $(501 \times 501\,\mathrm{px}^2)$ of a walnut (left), the absolute error of its CT reconstruction (top) and pixel-wise uncertainty (bottom).

---

[*] Equal contribution.

In recent years, deep-learning based approaches have achieved outstanding performance on a wide variety of tomographic problems (Arridge et al., 2019; Ongie et al., 2020; Wang et al., 2020). Most deep learning methods are supervised; they rely on large volumes of paired training data. Alas, these often fail to generalise out-of-distribution (Antun et al., 2020); small deviations from the distribution of the training data can lead to severe reconstruction artefacts. Pathologies of this sort call for both unsupervised deep learning methods—free from training data and thus mitigating hallucinatory artefacts (Bora et al., 2017; Heckel & Hand, 2019; Tölle et al., 2021)—and uncertainty quantification (Kompa et al., 2021; Vasconcelos et al., 2022)—informing the user about (un)reliability in reconstructions.

We focus on the deep image prior (DIP), perhaps the most widely adopted unsupervised deep learning approach (Ulyanov et al., 2018). DIP regularises the reconstructed image $\hat{x}$ by reparametrising it as the output of a deep convolutional neural network (CNN). It does not require paired training data, relying solely on the structural biases induced by the CNN architecture. The DIP has proven effective on tasks ranging from denoising and deblurring to challenging tomographic reconstructions (Liu et al., 2019; Baguer et al., 2020; Knopp & Grosser, 2022; Darestani & Heckel, 2021; Gong et al., 2019; Cui et al., 2021; Barutcu et al., 2022). Nonetheless, the DIP only provides point reconstructions without uncertainty estimates.

In this work, we equip DIP reconstructions with reliable uncertainty estimates, which is an under-explored topic. In literature, there are two notable probabilistic reformulations of the DIP (Cheng et al., 2019; Tölle et al., 2021), but their focus is on preventing overfitting rather than accurately estimating uncertainty. Distinctly from these, we only estimate the uncertainty associated with a specific reconstruction, instead of characterising a full posterior over all candidate images. We achieve this by computing Gaussian-linear model type error-bars for a local linearisation of the DIP around its mode (Mackay, 1992; Khan et al., 2019; Immer et al., 2021b), and refer to the method as *linearised* DIP. Linearised approaches have recently provided state-of-the-art uncertainty estimates for supervised deep learning models (Daxberger et al., 2021b). Unfortunately, the total variation (TV) regulariser, ubiquitous in CT reconstruction, makes inference in the linearised DIP intractable and it does not lend itself to standard Laplace (i.e. local Gaussian) approximations (Helin et al., 2022). We tackle this issue using predictive complexity prior (PredCP) (Nalisnick et al., 2021) to construct covariance kernels that induce properties similar to that of the TV prior while preserving Gaussian-linear conjugacy. Finally, we discuss several techniques to scale the method to large DIP networks and high-resolution 2D images.

We showcase our approach on high-resolution CT reconstructions of real-measured 2D $\mu$CT projection data, cf. fig. 1. Empirically, the method's pixel-wise uncertainty estimates predict reconstruction errors more accurately than existing approaches to uncertainty estimation with the DIP. This is not at the expense of accuracy in reconstruction: the reconstruction obtained using the standard regularised DIP method (Baguer et al., 2020) is preserved as the predictive mean, ensuring compatibility with advancements in DIP research.

The contributions of this work can be summarised as follows.

- We propose a novel approach to bestow reconstructions from the TV-regularised DIP with uncertainty estimates, by constructing a local linear model by linearising the DIP around its optimised reconstruction and providing the model's error-bars as a surrogate for those of the DIP.

- We give an efficient implementation of the method, scaling up to high-resolution $\mu$CT data, and yielding far more accurate uncertainty estimation than existing probabilistic formulations of the DIP.

The rest of this paper is organised as follows. Section 2 provides an extended discussion of the related work. Section 3 recalls preliminaries for the linearised DIP. Section 4 discusses the design of a tractable Gaussian prior mimicking the TV prior. Section 5 and section 6 present the linearised DIP and its efficient implementation. Section 7 presents the experimental investigations on synthetic and real-measured high-resolution $\mu$CT data. Section 8 concludes the article. Fully detailed derivations and additional experimental results are given in the supplementary material (SM).

Since this paper's first appearance, the proposed method was used by Barbano et al. (2022b) to actively select X-ray scanning angles, resulting in a 30% reduction in angles needed to obtain a given reconstruction PSNR, and extended by Antoran et al. (2023), scaling it to larger problems by drawing samples with SGD.

## 2 Related Work

### 2.1 Advances in the deep image prior

Since its introduction by Ulyanov et al. (2018; 2020), the DIP has been improved with early stopping (Wang et al., 2021), TV regularisation (Liu et al., 2019; Baguer et al., 2020) and pretraining (Barbano et al., 2022c; Knopp & Grosser, 2022; Barbano et al., 2023). We build upon these recent advancements by providing a scalable method to estimate the error-bars of DIP's reconstructions. Obtaining reliable uncertainty estimates for DIP reconstructions is a relatively unexplored topic. Building upon Garriga-Alonso et al. (2019) and Novak et al. (2019), Cheng et al. (2019) show that in the infinite-channel limit, the DIP converges to a Gaussian process (GP). In the finite-channel regime, the authors approximate the posterior distribution over the DIP's parameters with stochastic gradient Langevin dynamics (SGLD) (Welling & Teh, 2011). Laves et al. (2020) and Tölle et al. (2021) use factorised Gaussian variational inference (Blundell et al., 2015) and MC dropout (Hron et al., 2018; Vasconcelos et al., 2022), respectively. These probabilistic treatments of DIP primarily aim to prevent overfitting, as opposed to accurately estimating uncertainty. While they can deliver uncertainty estimates, their quality tends to be poor. In fact, obtaining reliable uncertainty estimates from deep-learning based approaches, like the DIP, largely remains a challenging open problem (Antorán, 2019; Snoek et al., 2019; Ashukha et al., 2020; Foong et al., 2020; Barbano et al., 2022a; Antorán et al., 2020). In the present work, we obtain uncertainty estimation by performing Bayesian inference with respect to the DIP model locally linearised around its optimised parameters. This is distinct from the aforementioned approaches in that we only model a local mode of the posterior distribution.

### 2.2 Bayesian inference in linearised neural networks

The Laplace method is first applied to deep learning in (Mackay, 1992). It has seen a recent popularisation as the best performing approach when it comes to Bayesian reasoning with neural networks (Daxberger et al., 2021b;a). Specifically, Khan et al. (2019) and Immer et al. (2021b) show that the linearization step improves the quality of uncertainty estimates. Immer et al. (2021a), Antorán et al. (2022) and Antorán et al. (2022) explore the linear model's evidence for model selection. Daxberger et al. (2021b) and Maddox et al. (2021) introduce subnetwork and finite differences approaches, respectively, for scalable inference with linearised models. Inference in the linearised model is highly attractive compared to alternative approaches because it is post-hoc and it preserves the reconstruction obtained through the DIP optimisation as the predictive mean. This line of work is also related to the neural tangent kernel (Jacot et al., 2018; Lee et al., 2019; Novak et al., 2020), in which NNs are linearised at initialisation.

## 3 Preliminaries

### 3.1 Total variation regularisation

The imaging problem given in eq. (1) admits multiple solutions consistent with the observation $y$. Thus, regularisation is needed for stable reconstruction. Total variation (TV) is perhaps the most well established regulariser (Rudin et al., 1992; Chambolle et al., 2010). The anisotropic TV semi-norm of an image vector $x \in \mathbb{R}^{d_x}$ imposes an $L^1$ constraint on image gradients:

$$\mathrm{TV}(x) = \sum_{i,j} |X_{i,j} - X_{i+1,j}| + \sum_{i,j} |X_{i,j} - X_{i,j+1}|, \tag{2}$$

where $X \in \mathbb{R}^{h \times w}$ denotes the vector $x$ reshaped into an image of height $h$ by width $w$, and $d_x = h \cdot w$. This leads to the regularised reconstruction formulation

$$\hat{x} \in \operatorname*{argmin}_{x \in \mathbb{R}^{d_x}} \mathcal{L}(x) \quad \text{with} \quad \mathcal{L}(x) := \|Ax - y\|_2^2 + \lambda \mathrm{TV}(x), \tag{3}$$

where the hyperparameter $\lambda > 0$ determines the strength of the regularisation relative to the fit term.

### 3.2 Bayesian inference for inverse problems

The Bayesian framework provides a consistent approach to uncertainty estimation in imaging problems (Kaipio & Somersalo, 2005; Stuart, 2010; Seeger & Nickisch, 2011). The image to be recovered is treated as a random variable. Instead of finding a single best reconstruction $\hat{x}$, we aim to find a posterior distribution $p(x|y)$ that scores every candidate $x \in \mathbb{R}^{d_x}$ according to its agreement with the observation $y$ and prior belief $p(x)$. The loss in eq. (3) can be viewed as the negative log of an unnormalised posterior, i.e. $p(x|y) \propto \exp(-\mathcal{L}(x))$, and $\hat{x}$ as its mode, i.e. the *maximum a posteriori* (MAP) estimate. The least squares loss corresponds to a Gaussian likelihood $p(y|x) = \mathcal{N}(y; Ax, I)$ and the TV regulariser to a prior over images $p(x) \propto \exp(-\lambda \mathrm{TV}(x))$.

The posterior is obtained by updating the prior over images with the likelihood as

$$p(x|y) = p(y)^{-1} p(y|x) p(x),$$ 
(4)

for $p(y) = \int p(y|x) p(x) \mathrm{d}x$ the normalising constant, also known as the marginal likelihood (MLL). This latter quantity provides an objective for optimising hyperparameters, e.g. the regularisation strength $\lambda$. The presence of different reconstructions with high probability under the posterior indicates uncertainty.

Our work partially departs from this framework in that it *solely concerns itself with characterising plausible reconstructions around the mode $\hat{x}$* (Mackay, 1992). This has two key advantages, i) *tractability*: the likelihood induced by NN reconstructions is strongly multi-modal, and both analytically and computationally intractable. In contrast, the posterior for the local model is Gaussian; ii) *interpretablity*: even if we could obtain the full posterior, downstream stakeholders not versed in probability are likely to have little use for it. A single reconstruction and its pixel-wise uncertainty may be more interpretable to end-users (Bhatt et al., 2021).

### 3.3 The Deep Image Prior (DIP)

The DIP (Ulyanov et al., 2018; 2020) reparametrises the reconstructed image as the output of a CNN $x(\theta)$ with learnable parameters $\theta \in \mathbb{R}^{d_\theta}$ and a fixed input, which we have omitted from our notation for clarity. The DIP can be seen as a reparametrisation that provides a favourable structural bias towards natural images. Penalising the TV of the DIP's output avoids the need for early stopping and improves reconstruction fidelity (Liu et al., 2019; Baguer et al., 2020). The resulting optimisation problem is given by

$$\hat{\theta} \in \underset{\theta \in \mathbb{R}^{d_\theta}}{\operatorname{argmin}} \|Ax(\theta) - y\|_2^2 + \lambda \mathrm{TV}(x(\theta)),$$
(5)

and the recovered image is given by $\hat{x} = x(\hat{\theta})$. U-Net is the standard choice of CNN architecture (Ronneberger et al., 2015). Although the parameters $\theta$ must be optimised separately for each new measurement $y$, we follow (Barbano et al., 2022c; Knopp & Grosser, 2022) to reduce the cost with task-agnostic pretraining.

### 3.4 Bayesian inference with linearised neural networks

Adopting the DIP parametrisation of the reconstructed image, as in section 3.3, makes the Bayesian posterior in eq. (4) intractable. Instead, this work only characterises the uncertainty associated with a specific regularised reconstruction $\hat{x}$, obtained via eq. (5). To this end, we take a *tangent linear model* of the CNN $x(\theta)$ around its optimised parameters $\hat{\theta}$ (Mackay, 1992; Khan et al., 2019; Immer et al., 2021b),

$$h(\theta) \coloneqq x(\hat{\theta}) + J(\theta - \hat{\theta}),$$
(6)

where $J \coloneqq \frac{\partial x(\theta)}{\partial \theta}\big|_{\theta=\hat{\theta}} \in \mathbb{R}^{d_x \times d_\theta}$ is the Jacobian of the CNN function $x(\theta)$ with respect to its parameters $\theta$ evaluated at $\hat{\theta}$. We obtain error-bars for the DIP reconstruction $x(\hat{\theta})$ using $h(\theta)$. For Gaussian noise and a Gaussian prior on $\theta$, we have a conjugate setting; the posterior over the linearised model's reconstructions is a Gaussian $\mathcal{N}(x; x(\hat{\theta}), \Sigma_{x|y})$, and the marginal likelihood of the linearised model can be used to tune hyperparameters (Mackay, 1992; Immer et al., 2021a; Antorán et al., 2022; Antorán et al., 2022).

Computing both the posterior covariance $\Sigma_{x|y}$ and the marginal likelihood naively has cost $\mathcal{O}(d_\theta^3)$. For large U-Nets, this is impracticable (Daxberger et al., 2021b). In section 5 and section 6, we derive a dual approach with a cost $\mathcal{O}(d_y^3)$ and detail an efficient implementation. Furthermore, when using the (non-quadratic) TV regulariser, conjugacy is lost. Indeed, the TV regulariser does not admit a Laplace (quadratic) approximation.

## 4   The total variation as a conditionally Gaussian prior

First, we study the construction of tractable non-DIP-based priors for CT reconstruction. The gained understanding sheds insights into incorporating the TV-based priors into the linearised DIP framework. The regularised loss in eq. (3) can be interpreted as the negative log of an unnormalised posterior over reconstructions. In this context, the TV regulariser corresponds to the prior

$$p(x) = Z_\lambda^{-1} \exp(-\lambda \mathrm{TV}(x)), \tag{7}$$

where $Z_\lambda = \int \exp(-\lambda \mathrm{TV}(x))\,\mathrm{d}x$ is its normalisation constant (the prior is improper, since constant vectors are in the null space of the derivative operator). Working with the prior $p(x)$ is intractable since $Z_\lambda$ does not admit a closed form. The Laplace method, which consists of a locally quadratic approximation, does not solve the issue because the second derivative of the TV regulariser is zero everywhere it is defined.

To enforce local smoothness in the reconstruction, we construct a Gaussian prior $\mathcal{N}(x; \mu, \Sigma_{xx})$ with mean $\mu \in \mathbb{R}^{d_x}$ and covariance $\Sigma_{xx} \in \mathbb{R}^{d_x \times d_x}$ given by the Matern-$1/2$ kernel

$$[\Sigma_{xx}]_{ij,i'j'} = \sigma^2 \exp\left(\frac{-\mathrm{d}(i - i', j - j')}{\ell}\right), \tag{8}$$

where $i, j$ index the spatial locations of pixels of $x$, as in eq. (2), and $\mathrm{d}(a, b) = \sqrt{a^2 + b^2}$. The hyperparameter $\sigma^2 \in \mathbb{R}^+$ informs the pixel amplitude while the lengthscale parameter $\ell \in \mathbb{R}^+$ determines the correlation strength between nearby pixels. The expected TV associated with our Gaussian prior is

$$\kappa := \mathbb{E}_{x \sim \mathcal{N}(\mu, \Sigma_{xx})}[\mathrm{TV}(x)] = c\sigma\sqrt{1 - \exp(-\ell^{-1})}, \tag{9}$$

with $c$ a constant. See appendix A for a derivation. Below we may omit the dependence of $\kappa$ on $(\ell, \sigma^2)$ from the notation. For fixed pixel amplitude $\sigma^2$, the expected reconstruction TV $\kappa$ is a bijection of the lengthscale $\ell$. We leverage this fact within the PredCP framework of Nalisnick et al. (2021) to construct a prior over $\ell$ that favours reconstructions with low expected TV

$$p(\ell) = \mathrm{Exp}(\kappa) \left|\partial\kappa/\partial\ell\right|, \tag{10}$$

where Exp is the density of the exponential distribution. The resulting hierarchical prior over images

$$x|\ell \sim \mathcal{N}(\mu, \Sigma_{xx}), \quad \ell \sim \mathrm{Exp}(\kappa) \left|\partial\kappa/\partial\ell\right| \tag{11}$$

is Gaussian for fixed $\ell$, and thus the prior is conditionally conjugate to Gaussian-linear likelihoods. Figure 2 shows agreement between samples, drawn with Hamiltonian Monte Carlo, from the described TV-PredCP prior and the intractable TV prior, both qualitatively and in terms of distribution over image TV. The TV prior produces samples with more correlated nearby pixel values than the factorised prior. The TV-PredCP prior captures this effect and produces even smoother samples, likely due to the presence of longer range correlation in the Matern-$1/2$ covariance.

## 5   The linearised DIP

In this section, we build a probabilistic model to characterise posterior reconstructions around $\hat{\theta}$, a mode of the regularised DIP objective (obtained using eq. (5)). Section 5.1 describes the construction of a linearised surrogate for the DIP reconstruction. Section 5.2 describes how to compute the surrogate model's error-bars and use them to augment the DIP reconstruction. Section 5.3 discusses how we include the effects of TV regularisation into the surrogate model. Finally, in section 5.4, we describe a strategy to choose the surrogate model's prior hyperparameters using a marginal likelihood objective.

### 5.1   From a prior over parameters to a prior over images

After training the DIP to an optimal TV-regularised setting $\hat{x} = x(\hat{\theta})$ using eq. (5), we linearise the network around $\hat{\theta}$ by applying eq. (6), and obtain the affine-in-$\theta$ function $h(\theta)$. The error-bars obtained from Bayesian

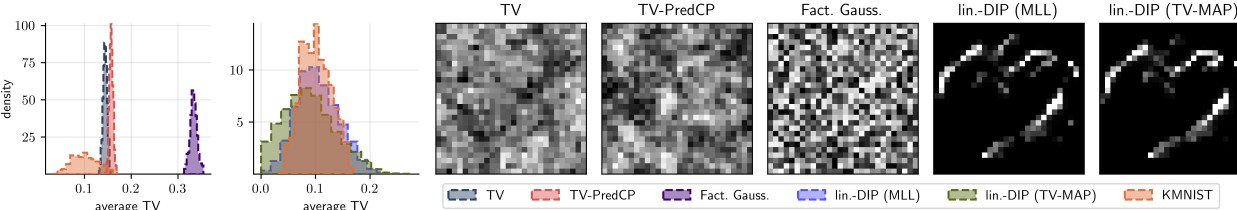

Figure 2: Samples from priors. From left to right. Plot 1 shows a histogram of the average sample TV reporting an overlap between the TV and TV-PredCP priors. The factorised Gaussian prior results in larger TV values. Plot 2 shows an analogous histogram using samples from the linearised DIP (lin.-DIP) fitted to a KMNIST image, where the hyperparameters ($\ell$, $\sigma^2$) have been optimised both with and without the TV-PredCP term. The TV-PredCP term in the DIP hyperparameter optimisation leads to smoother samples with less artefacts. Plots 3-5 show samples from the TV, TV-PredCP, and factorised Gaussian priors proposed in section 4, drawn using Hamiltonian Monte Carlo (HMC). Plots 6-7 show prior samples from the linearised DIP, which produces samples containing the structure of the KMNIST image used to train the network.

inference with $h(\theta)$ will tell us about the uncertainty in $\hat{x}$. To this end, consider the hierarchical model,

$$y|\theta \sim \mathcal{N}(\mathrm{A}h(\theta), \sigma_y^2 \mathrm{I}), \quad \theta|\ell \sim \mathcal{N}(0, \Sigma_{\theta\theta}(\ell)), \quad \ell \sim p(\ell) \quad \text{with} \quad h(\theta) \coloneqq x(\hat{\theta}) + J(\theta - \hat{\theta}), \tag{12}$$

where we place a Gaussian prior over the parameters $\theta$ that, in turn, depends on the lengthscale $\ell$. Conditioned on the value of $\ell$, this is a conjugate Gaussian-linear model and thus the posterior distribution over $\theta$ has a closed Gaussian form. Learning the lengthscale $\ell$ will allow us to incorporate TV constraints into the computed error-bars, cf. section 5.3. We have introduced the noise variance $\sigma_y^2$ as an additional hyperparameter which we will learn using the marginal likelihood (cf. section 5.4).

To provide intuition about the linearised model, we push samples from $\theta \sim \mathcal{N}(\theta; 0, \Sigma_{\theta\theta})$, through $h$. The resulting reconstruction samples are drawn from a Gaussian distribution with covariance $\Sigma_{xx} \in \mathbb{R}^{d_x \times d_x}$ given by $J\Sigma_{\theta\theta}J^\top$ and are shown in fig. 2. Here, the Jacobian $J$ introduces structure from the NN function around the linearisation point $\hat{\theta}$. It introduces features from the KMNIST character that the DIP was trained on.

## 5.2 Efficient posterior predictive computation

We augment the DIP reconstruction $\hat{x}$ with Gaussian predictive error-bars computed with the linearised model $h$ described in eq. (12), yielding $\mathcal{N}(x; \hat{x}, \Sigma_{x|y})$. The posterior covariance $\Sigma_{x|y}$ is given by

$$\Sigma_{x|y} = J(\sigma_y^{-2} J^\top \mathrm{A}^\top \mathrm{A} J + \Sigma_{\theta\theta}^{-1})^{-1} J^\top = \Sigma_{xx} - \Sigma_{xy} \Sigma_{yy}^{-1} \Sigma_{xy}^\top, \tag{13}$$

which is derived in appendix B. Here, $\Sigma_{xx} = J\Sigma_{\theta\theta}J^\top$, $\Sigma_{xy} = \Sigma_{xx}A^\top$ and $\Sigma_{yy} = A\Sigma_{xx}A^\top + \sigma_y^2 I$. The constant-in-$\theta$ terms in $h$ do not affect the uncertainty estimates, and thus the error-bars match those of the simple linear model $J\theta$. Importantly, eq. (13) depends on the inverse of the observation space covariance $\Sigma_{yy}^{-1}$, as opposed to the covariance over reconstructions, or parameters. Equation (13) scales as $\mathcal{O}(d_x d_y^2)$ as opposed to $\mathcal{O}(d_x^3)$ or $\mathcal{O}(d_\theta^3)$ for the more-standard-in-the-literature output (reconstruction) space or parameter space approaches, respectively (Immer et al., 2021b; Daxberger et al., 2021a).

## 5.3 Incorporating TV-smoothness into our model as a prior

We impose constraints on $h$'s error-bars, such that the model only considers low TV reconstructions as plausible. For this, we place a block-diagonal Matern-1/2 covariance Gaussian prior on the linearised model's weights, similarly to Fortuin et al. (2021). We introduce dependencies between parameters in the same CNN convolutional filter as

$$[\Sigma_{\theta\theta}]_{kij,k'i'j'} = \sigma_d^2 \exp\left(\frac{-\mathrm{d}(i - i', j - j')}{\ell_d}\right)\delta_{kk'}, \tag{14}$$

where $k$ indexes the convolutional filters in the CNN, $\delta_{kk'}$ denotes Kronecker symbol, and $(i, j)$ index the spatial locations of specific parameters within a filter. The lengthscale $\ell_d$ regulates the filter smoothness.

Intuitively, an image generated from convolutions with smoother filters will present lower TV. Indeed, in appendix C we show a bijective relationship between this quantity and the filter lengthscale. The hyperparameter $\sigma_d^2$ determines the marginal prior variance. Both parameters are defined per architectural block $d \in \{1, 2, \ldots, D\}$ in the U-Net and we write $\ell = [\ell_1, \ell_2, \ldots, \ell_D]$ and $\sigma^2 = [\sigma_1^2, \sigma_2^2, \ldots, \sigma_D^2]$. The chosen U-Net architecture is fully convolutional and thus eq. (14) applies to all parameters, reducing to a diagonal covariance for $1 \times 1$ convolutions. A U-Net diagram highlighting these prior blocks is in fig. 3.

To enforce TV-smoothness, we adopt the strategy given in section 4. Since choosing a large $\ell$ enforces smoothness in the output, a prior placed over the filter lengthscales $\ell$ can act as a surrogate for the TV prior.

To make this connection explicit, we construct a TV-PredCP (Nalisnick et al., 2021)

$$p(\ell) = \prod_{d=1}^{D} p(\ell_d) = \prod_{d=1}^{D} \text{Exp}(\kappa_d) \left| \frac{\partial \kappa_d}{\partial \ell_d} \right|, \tag{15}$$

$$\text{with } \kappa_d := \mathbb{E}_{\theta \sim \mathcal{N}(\hat{\theta}_d, \Sigma_{\theta_d \theta_d}) \prod_{i=1, i \neq d}^{D} \delta(\theta_i - \hat{\theta}_i)} \left[ \lambda \text{TV}(h(\theta)) \right] \tag{16}$$

being the expected TV of the CNN output over the prior uncertainty in the parameters of block $d$ when all other entries of $\theta$ are fixed to $\hat{\theta}$.

We relate the expected TV $\kappa_d$ to the filter lengthscale $\ell_d$ via the change of variables formula. The independence across blocks of $p(\ell)$ ensures dimensionality preservation, formally needed in changing variables. It follows from the triangle inequality that $\sum_d \kappa_d$ is an upper bound on the expectation under the distribution $\mathbb{E}_{\theta \sim \mathcal{N}(\hat{\theta}, \Sigma_{\theta \theta})}[\text{TV}(h(\theta))]$, cf. appendix C.

Note that eq. (15) can be computed analytically. However, its direct computation is costly and we instead rely on numerical methods described in section 6. In fig. 2 (cf. plot 2 and plots 6-7), we show samples from $\mathcal{N}(x; 0, \Sigma_{xx})$ where $\ell$ is chosen using the marginal likelihood with TV-PredCP constraints (cf. also section 5.4). Incorporating the TV-PredCP leads to smoother samples with less discontinuities.

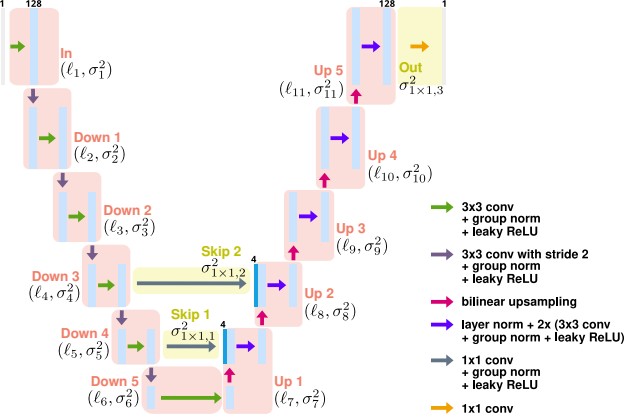

Figure 3: A schematic of the U-Net architecture used in the numerical experiments on Walnut data (see section 7.2). For KMNIST, we use a reduced, 3-scale U-Net without group norm layers (see fig. 20). Each light-blue rectangle corresponds to a multi-channel feature map. We highlight the architectural components corresponding to each block $1, \ldots, D$ for which a separate prior is defined with red and yellow boxes.

### 5.4 Type-II MAP learning of hyperparameters

The calibration of the predictive Gaussian error-bars depends on the choice of the hyperparameters $(\sigma_y^2, \sigma^2, \ell)$ of the hierarchical model in eq. (12) (Antorán et al., 2022). For a given $\ell$, Gaussian-linear conjugacy yields a closed form marginal likelihood objective to learn the hyperparameters. In turn, to learn $\ell$, we combine the above objective with the TV-PredCP's log-density, which acts as a regulariser. The resulting expression resembles a Type-II MAP (Rasmussen & Williams, 2005) objective

$$\log p(y|\ell; \sigma_y^2, \sigma^2) + \log p(\ell; \sigma^2) \approx$$

$$-\frac{1}{2} \sigma_y^{-2} ||y - \mathbf{A}x(\hat{\theta})||_2^2 - \frac{1}{2} \hat{\theta}_h^\top \Sigma_{\theta \theta}^{-1}(\ell, \sigma^2) \hat{\theta}_h - \frac{1}{2} \log |\Sigma_{yy}| - \sum_{d=1}^{D} \kappa_d(\ell, \sigma^2) + \log \left| \frac{\partial \kappa_d(\ell, \sigma^2)}{\partial \ell_d} \right| + B, \tag{17}$$

where $B$ is independent of $(\sigma_y^2, \sigma^2, \ell)$ and the vector $\hat{\theta}_h \in \mathbb{R}^{d_\theta}$ is the posterior mean of the linear model's parameters. See appendix B for the detailed derivation. The bottleneck in evaluating eq. (17) is the log-determinant $\log |\Sigma_{yy}|$ of $\Sigma_{yy}$, which has a cost $\mathcal{O}(d_y^3)$. We go on to describe scalable ways to approximate the log-determinant and other costly quantities required for prediction.

# 6 Towards scalable computation

In a typical tomography setting, the dimensionality $d_x$ of the image $\hat{x}$ and $d_y$ of the observation $y$ can be large, e.g. $d_x > 1\mathrm{e}5$ and $d_y > 1\mathrm{e}3$. Thus holding the input space covariance matrices (e.g. $\Sigma_{xx}$ and $\Sigma_{x|y}$) in memory is infeasible. The latter complicates computing $\log|\Sigma_{yy}|$ in eq. (17) (or its gradients), and its inverse in $\Sigma_{x|y}$, cf. eq. (13), which scale as $\mathcal{O}(d_y^3)$ and $\mathcal{O}(d_x d_y^2)$, respectively. To scale the approach, we only access Jacobian and covariance matrices through matrix–vector products (*matvecs*), i.e. products resembling $v_x^\top \Sigma_{xx}$ and $v_y^\top \Sigma_{yy}$ for $v_x \in \mathbb{R}^{d_x}$ and $v_y \in \mathbb{R}^{d_y}$. We compute $v_y \in \mathbb{R}^{d_y}$ through successive matvecs as

$$v_y^\top \Sigma_{yy} = v_y^\top (\mathrm{A}J\Sigma_{\theta\theta}J^\top \mathrm{A}^\top + \sigma_y^2 \mathrm{I}), \tag{18}$$

and we compute $v_x^\top \Sigma_{xx}$ similarly. We compute Jacobian vector products $v_x^\top J^\top$ for $v_\theta \in \mathbb{R}^{d_\theta}$ using forward mode automatic differentiation (AD) and $v_x^\top J$ using backward mode AD, both with the `functorch` library (He & Zou, 2021). We compute products with $\Sigma_{\theta\theta}$ by exploiting its block diagonal structure. All these operations can be batched using modern numerical libraries and GPUs.

## 6.1 Conjugate gradient log-determinant gradients

For the Type-II MAP optimisation in eq. (17), we estimate the gradients of $\log|\Sigma_{yy}|$ with respect to the parameters of interest $\phi$ using the stochastic trace estimator (Gibbs & MacKay, 1996; Gardner et al., 2018)

$$\frac{\partial \log|\Sigma_{yy}|}{\partial \phi} = \mathrm{Tr}\left(\Sigma_{yy}^{-1}\frac{\partial \Sigma_{yy}}{\partial \phi}\right) = \mathbb{E}_{v \sim \mathcal{N}(0,P)}\left[v^\top \Sigma_{yy}^{-1}\frac{\partial \Sigma_{yy}}{\partial \phi}P^{-1}v\right], \tag{19}$$

where $P$ is a preconditioner matrix. We approximately solve the linear system $v^\top \Sigma_{yy}^{-1}$ for batches of probe vectors $v$ using the `GPyTorch` preconditioned conjugate gradient (PCG) implementation (Dong et al., 2017).

The preconditioner $P$ is constructed using $r$-rank randomised SVD, by approximating $\mathrm{A}J\Sigma_{\theta\theta}J^\top \mathrm{A}^\top$ as $\tilde{U}\tilde{\Lambda}\tilde{U}^\top$, using a randomised eigendecomposition algorithm (Halko et al., 2011; Martinsson & Tropp, 2020) with $\tilde{U} \in \mathbb{R}^{d_y \times r}$ and $r = 200 \ll d_y$. The algorithm is described in detail in appendix E. Since $P$ depends on the hyperparameters $\phi$, we interweave the updates of $P$ with the optimisation of eq. (17).

## 6.2 Ancestral sampling for TV-PredCP optimisation

For large images, exact evaluation of the expected TV with eq. (16) is intractable. Instead, we estimate the gradient of $\kappa_d$ with respect to $\phi = (\sigma^2, \ell)$ using a Monte-Carlo approximation

$$\frac{\partial \kappa_d}{\partial \phi} = \mathbb{E}_{\theta_d \sim \mathcal{N}(\hat{\theta}_d, \Sigma_{\theta_d\theta_d})}\left[\frac{\partial \mathrm{TV}(x)}{\partial x}J_d\frac{\partial \theta_d}{\partial \phi}\right], \tag{20}$$

where $J_d = \frac{\partial x(\theta)}{\partial \theta_d}\big|_{\theta_d = \hat{\theta}_d}$, $\frac{\partial \mathrm{TV}(x)}{\partial x}$ is evaluated at the sample $x = J_d\theta_d$ and $\frac{\partial \theta_d}{\partial \phi}$ is the reparametrisation gradient for $\theta_d$, a prior sample of the weights of CNN block $d$. Since the second derivative of the TV semi-norm is almost everywhere zero, the gradient for the change of variables volume ratio is

$$\frac{\partial^2 \kappa_d}{\partial \phi^2} = \mathbb{E}_{\theta_d \sim \mathcal{N}(\hat{\theta}_d, \Sigma_{\theta_d\theta_d})}\left[\frac{\partial \mathrm{TV}(x)}{\partial \phi}J_d\frac{\partial^2 \theta_d}{\partial \phi^2}\right]. \tag{21}$$

## 6.3 Posterior covariance matrix estimation by sampling

The covariance matrix $\Sigma_{x|y}$ is too large to fit into memory for high-resolution tomographic reconstructions. Instead, we follow Wilson et al. (2021) in drawing samples from $\mathcal{N}(x; 0, \Sigma_{x|y})$ via Matheron's rule

$$x_{x|y} = x_0 + \Sigma_{xy}\Sigma_{yy}^{-1}(\epsilon - \mathrm{A}x_0); \quad x_0 = J\theta_0; \quad \theta_0 \sim \mathcal{N}(0, \Sigma_{\theta\theta}); \quad \epsilon \sim \mathcal{N}(0, \sigma_y^2 \mathrm{I}). \tag{22}$$

The biggest cost lies in constructing $\Sigma_{yy}$, which is achieved by applying eq. (18) to the standard basis vectors $\Sigma_{yy} = [e_1, e_2, \dots e_{d_y}]^\top \Sigma_{yy}$. We then perform its Cholesky factorisation as an intermediate step towards matrix

inversion, both relatively costly operations. Fortunately, we only have to repeat these once, after which the sampling step in eq. (22) can be evaluated cheaply. Alternatively, as in eq. (19), we can compute the solution of the linear system, $\Sigma_{yy}^{-1} v_y$ for any $v_y$ via PCG, without explicitly assembling (and thus storing in memory) the measurement covariance matrix, or computing its Cholesky factorisation. This approach allows us to scale the sampling operation to large measurement spaces, where the matrix $\Sigma_{yy}$ may not fit in memory.

Since only nearby pixels of the predictions are expected to be correlated, we estimate cross covariances for patches of only up to $10 \times 10$ adjacent pixels. Using larger patches yields no improvements. We use the stabilised formulation of Maddox et al. (2019): $\hat{\Sigma}_{x|y} = \frac{1}{2k}[\sum_{j=1}^{k} \text{diag}(x_j)^2 + x_j x_j^\top]$ for $(x_j)_{j=1}^{k}$ samples from the posterior predictive distribution over a patch. Note that the samples from eq. (22) are zero mean.

### 6.4 Faster low-rank Jacobian matvecs

Table 1 shows that the Jacobian matvecs—implemented through forward and backward mode AD— required for sampling from the posterior predictive (that is $2\times v_\theta^\top J^\top$ and $1\times v_x^\top J$) take $\approx 100\%$ of this step's computation time (2.4 h). To accelerate sampling, we construct a low-rank approximation of the Jacobian $\tilde{J}$, which we store in memory. We compute $v_\theta^\top \tilde{J}^\top$ and $v_x^\top \tilde{J}$ via matvec, as opposed to AD. This allows for fast approximation of $v_y^\top \Sigma_{yy}$ by substituting $\tilde{J}$ into eq. (18). This brings the time needed for sampling from the posterior predictive down from 2.4 hours to less than a minute. We construct $\tilde{J}$ similarly to the low-rank preconditioner $P$ (see section 6.1 and appendix E). That is, following

Table 1: Wall-clock time on an A100 GPU for the different steps of our algorithm when applied to high-resolution CT (details in section 7.2). Computations reported below the dotted line are in double precision. The time taken by Jacobian matvecs during sampling is given in parenthesis.

| | wall-clock time |
|---|---|
| DIP optim. (after pretraining (Barbano et al. 2022c)) | <0.1 h |
| Hyperparam. optim. (MLL) | 26.2 h |
| Hyperparam. optim. (TV-MAP) | 35.4 h |
| Assemble $\Sigma_{yy}$ | 2.7 h |
| Draw 4096 posterior samples | 2.4 h |
| - (Evaluate 4096 times $2\times v_\theta^\top J^\top + 1\times v_x^\top J$) | 2.4 h |
| Draw 4096 posterior samples ($\tilde{J}$ & PCG) | 0.3 h |
| - (Evaluate 4096 times $2\times v_\theta^\top \tilde{J}^\top + 1\times v_x^\top \tilde{J}$) | < 0.1 min |

Halko et al. (2011), we build a rank-$r$ approximation to $J$, by accessing only to matvecs with $J$ and $J^\top$. While offering a well-calibrated alternative to uncertainty quantification within the DIP framework, it incurs computational overhead (see table 1) when compared to MC dropout, which only require a forward pass through the network to generate a single sample.

Algorithm 1 summarises image reconstruction and uncertainty estimation with the linearised DIP.

---

**Algorithm 1:** Linearised deep image prior (lin.-DIP) inference

**Inputs:** noisy measurements $y$, a CNN $x(\cdot)$, probabilistic model's hyperparameters, whether to use fast approximate posterior sampling *fast_sampling*

1   $\hat{\theta} \leftarrow$ `fit_DIP`$(y, x(\theta))$ // by minimising eq. (5)

2   $\hat{\theta}_h \leftarrow$ `find_linearised_MAP`$(y, x(\hat{\theta}))$ // using Algorithm 1 from Antorán et al. (2022)

3   $\sigma_y^2, \{\sigma_d^2, \ell_d\}_{d=1}^{D} \leftarrow$ `optimise_hyperparams`$(y, x(\hat{\theta}), \hat{\theta}_h)$ // by maximising eq. (17) with
     estimators eqs. (19) to (21) and solving linear systems with PCG

4   **if** *not fast_sampling* **then**

5      $\Sigma_{yy} \leftarrow$ `assemble_covariance`$(x(\hat{\theta}), \sigma_y^2, \{\sigma_d^2, \ell_d\}_{d=1}^{D})$ // by applying eq. (18) to rows of $\text{I}_{d_y}$

6      $\hat{\Sigma}_{x|y} \leftarrow$ `posterior_sampling`$(x(\hat{\theta}), \{\sigma_d^2, \ell_d\}_{d=1}^{D}, \Sigma_{yy})$ // using eq. (22)

7   **else**

8      $\tilde{J} \leftarrow$ `construct_lowrank_Jacobian`$(x(\hat{\theta}))$ // by randomised SVD, cf. section 6.4

9      $\hat{\Sigma}_{x|y} \leftarrow$ `fast_sampling`$(x(\hat{\theta}), \sigma_y^2, \{\sigma_d^2, \ell_d\}_{d=1}^{D}, \tilde{J})$ // using eq. (22) with $\tilde{J}$ and PCG

**Output:** mean reconstruction $x(\hat{\theta})$, posterior covariance estimate $\hat{\Sigma}_{x|y}$

---

## 7 Experiments

Here, we experimentally evaluate: i) the properties of the models and priors discussed in sections 4 and 5, and whether they lead to accurate reconstructions and calibrated uncertainty; ii) the fidelity of the approximations described in section 6; and iii) the performance of the proposed method linearised DIP (lin.-DIP) relative to the previous MC dropout (MCDO) based probabilistic formulation of DIP (Laves et al., 2020). We attempted to include DIP-SGLD (Cheng et al., 2019) in our analysis, but were unable to get the method to produce competitive results on tomographic reconstruction problems. For each individual image to be reconstructed, we employ the following linearised DIP inference procedure: i) optimise the DIP weights via eq. (5), obtaining $\hat{x} = x(\hat{\theta})$; ii) optimise prior hyperparameters ($\sigma_y^2$, $\ell$, $\sigma^2$) via eq. (17); iii) assemble and Cholesky decompose $\Sigma_{yy}$ with eq. (18) (this step can be accelerated using approximate methods sections 6.3 and 6.4); iv) compute posterior covariance matrices either via eq. (13), or estimate them via eq. (22); cf. Algorithm 1.

### 7.1 Small scale ablation analysis: reconstruction of KMNIST digits

The initial analysis uses simulated CT data obtained by applying eq. (1) to 50 images from the test set of the Kuzushiji-MNIST (KMNIST) dataset: $28 \times 28$ ($d_x = 784$) grayscale images of Hiragana characters (Clanuwat et al., 2018). We choose the noise standard deviation to be either 5% or 10% of the mean of A$x$, denoted as $\eta(5\%)$ or $\eta(10\%)$. The forward map A is a discrete Radon transform, assembled via ODL (Adler et al., 2017). We use a U-Net with 3 scales and 76905 parameters (a down-sized net compared to the one in fig. 3).

#### 7.1.1 Comparing linearised DIP with network-free priors

We first evaluate the priors in section 4, i.e. TV prior, TV-PredCP with a Matern-½ kernel, and a factorised Gaussian prior, and perform inference in the setting where the map A collects 5 angles ($d_y = 205$) sampled uniformly from 0° to 180° and is applied to 50 KMNIST test set images. Here, 10% noise is added. This results in a very ill-posed reconstruction problem, maximising the relevance of the prior. We select the $\sigma_y^2$ and $\lambda$ hyperparameters for the factorised Gaussian prior and the TV prior

Table 2: Quantitative results for inference with the different priors introduced in section 4. We report both the PSNR of $\mathbb{E}[x|y]$, which denotes the posterior mean reconstruction, and the PSNR of $\hat{x}$, which denotes the posterior mode found through optimisation.

|  | log-likelihood | $\mathbb{E}[x|y]$ | $\hat{x}$ |
|---|---|---|---|
| Fact. Gauss. | $0.30 \pm 0.17$ | $16.15 \pm 0.38$ | $14.89 \pm 0.38$ |
| TV | $0.49 \pm 0.14$ | $16.32 \pm 0.38$ | $16.29 \pm 0.41$ |
| TV-PredCP | $\mathbf{0.65 \pm 0.12}$ | $\mathbf{16.55 \pm 0.39}$ | $\mathbf{17.48 \pm 0.39}$ |
| lin.-DIP (MLL) | $1.63 \pm 0.08$ | – | $19.46 \pm 0.52$ |
| lin.-DIP (TV-MAP) | $1.63 \pm 0.09$ | – | $19.46 \pm 0.52$ |

respectively such that the posterior mean's PSNR is maximised across a validation set of 10 images from the KMNIST training set. We keep the choice of $\sigma_y^2$ and $\lambda$ hyperparameters from the first two models for our experiments with the third model: Matern-½ with TV-PredCP prior over $\ell$. For all priors, we perform inference with the NUTS HMC sampler. We run 5 independent chains for each image. We burn these in for $3 \times 10^3$ steps each and then proceed to draw $10^4$ samples with a thinning factor of 2.

We evaluate test log-likelihood using Gaussian Kernel Density Estimation (KDE) (Silverman, 1986). The kernel bandwidth is chosen using cross-validation on 10 images from the training set. The results in table 2 show that the TV-PredCP performs best in terms of the test log-likelihood and both posterior mean and posterior mode PSNR, followed by the TV and then the factorised Gaussian. This is somewhat surprising considering that this prior was designed as an approximation to the intractable TV prior. We hypothesise that this may be due to the Matern model allowing for faster transitions in the image than the TV prior, while still capturing local correlations, as shown qualitatively in fig. 2. This property may be well-suited to the KMNIST datasets, where most pixels either present large amplitudes or are close to 0. DIP-based predictions provide 2dB higher PSNR reconstructions than the non-DIP based priors, thus linearised DIP handily obtains a better test log-likelihood than the more-traditional methods.

#### 7.1.2 Comparing calibration with DIP uncertainty quantification baselines

Using KMNIST, we construct test cases of different ill-posedness by simulating the observation $y$ with four different angle sub-sampling settings for the linear operator A: 30 ($d_y=1230$), 20 ($d_y=820$), 10 ($d_y=410$) and 5 ($d_y=205$) angles are taken uniformly from the range 0° to 180°. We consider two noise configurations by

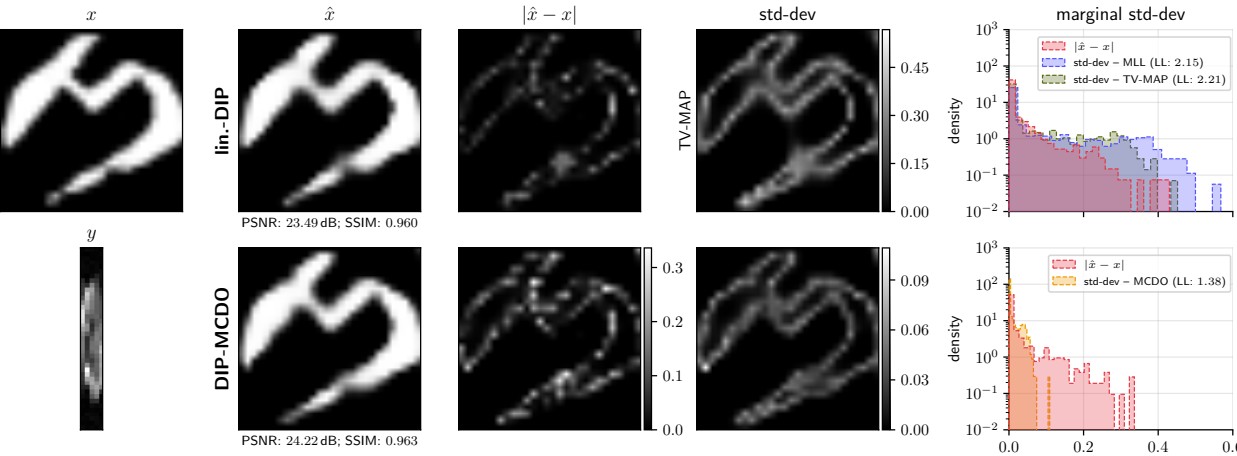

Figure 4: Exemplary character recovered from $y$ (using 5 angles and $\eta(5\%)$) with lin.-DIP and DIP-MCDO along with respective uncertainty estimates. lin.-DIP provides vastly improved uncertainty calibration. For lin.-DIP, the colour-map is shared between $|\hat{x} - x|$ and std-dev, and TV-MAP refers to Type-II MAP optimisation of hyperparameters.

adding either 5% or 10% noise to the exact data $Ax$. We evaluate all DIP-based methods using the same 50 randomly chosen KMNIST test set images. To ensure a best-case showing of the methods, we choose appropriate hyperparameters for each number of angles and white noise percentage setting by applying grid-search cross-validation, using 50 images from the KMNIST training dataset. Specifically, we tune the TV strength $\lambda$ and the number of optimisation iterations for the DIP. Due to the reduced image size, we apply linearised DIP as in section 5, without approximate computations. As an ablation study, we include additional baselines: linearised DIP without the TV-PredCP prior over hyperparameters (labelled MLL), and DIP reconstruction with a simple Gaussian noise model consisting of the back-projected observation noise $\mathcal{N}(x; \hat{x}, \sigma_A^2 I)$, with $\sigma_A^2 = \sigma_y^2 \mathrm{Tr}((A^\top A)^\dagger) d_x^{-1}$ where $\sigma_y^2 = 1$ (labelled $\sigma_y^2 = 1$). Note that non-dropout methods share the same DIP parameters $\hat{\theta}$, and thus the same mean reconstruction. Hence, higher values in log-density indicate better uncertainty calibration, i.e. the predictive standard deviation better matches the empirical reconstruction error. DIP-MCDO does not provide an explicit likelihood function over the reconstructed image. We model its uncertainty with a Gaussian predictive distribution with covariance estimated from $2^{14}$ samples. MNIST images are quantised to 256 bins, but our models make predictions over continuous pixel values. Thus, we simulate a de-quantisation of KMNIST images by adding a noise jitter term of variance approximately matching that of a uniform distribution over the quantisation step (Hoogeboom et al., 2020).

Table 3: Mean and std-err of test log-likelihood computed over 50 KMNIST test images.

| $\eta$ (5%) | #angles: 5 | 10 | 20 | 30 | $\eta$ (10%) | #angles: 5 | 10 | 20 | 30 |
|---|---|---|---|---|---|---|---|---|---|
| DIP ($\sigma_y^2 = 1$) | $0.68 \pm 0.14$ | $1.57 \pm 0.02$ | $1.85 \pm 0.02$ | $2.02 \pm 0.02$ | DIP ($\sigma_y^2 = 1$) | $0.27 \pm 0.17$ | $1.31 \pm 0.04$ | $1.62 \pm 0.03$ | $1.76 \pm 0.04$ |
| DIP-MCDO | $0.74 \pm 0.13$ | $1.60 \pm 0.02$ | $1.87 \pm 0.02$ | $2.05 \pm 0.02$ | DIP-MCDO | $0.42 \pm 0.14$ | $1.39 \pm 0.04$ | $1.70 \pm 0.03$ | $1.85 \pm 0.04$ |
| lin.-DIP (MLL) | $\mathbf{1.90 \pm 0.14}$ | $\mathbf{2.57 \pm 0.09}$ | $\mathbf{2.94 \pm 0.10}$ | $\mathbf{3.09 \pm 0.12}$ | lin.-DIP (MLL) | $\mathbf{1.63 \pm 0.08}$ | $\mathbf{2.11 \pm 0.07}$ | $\mathbf{2.43 \pm 0.07}$ | $\mathbf{2.59 \pm 0.08}$ |
| lin.-DIP (TV-MAP) | $\mathbf{1.88 \pm 0.15}$ | $\mathbf{2.59 \pm 0.10}$ | $\mathbf{2.96 \pm 0.10}$ | $\mathbf{3.11 \pm 0.12}$ | lin.-DIP (TV-MAP) | $\mathbf{1.63 \pm 0.09}$ | $\mathbf{2.13 \pm 0.07}$ | $\mathbf{2.45 \pm 0.08}$ | $\mathbf{2.61 \pm 0.08}$ |

Table 4: PSNR [dB] / SSIM of the reconstruction posterior mean, averaged over 50 KMNIST test images.

| $\eta$ (5%) | #angles: 5 | 10 | 20 | 30 | $\eta$ (10%) | #angles: 5 | 10 | 20 | 30 |
|---|---|---|---|---|---|---|---|---|---|
| DIP | $\mathbf{21.42}/\ \mathbf{0.890}$ | $27.92/\ \mathbf{0.977}$ | $31.21/\ \mathbf{0.988}$ | $32.93/\ \mathbf{0.991}$ | DIP | $\mathbf{19.46}/\ \mathbf{0.846}$ | $24.56/\ \mathbf{0.956}$ | $27.27/\ \mathbf{0.974}$ | $28.57/\ \mathbf{0.980}$ |
| DIP-MCDO | $20.95/0.882$ | $\mathbf{28.26}/\ \mathbf{0.977}$ | $\mathbf{31.65}/0.986$ | $\mathbf{33.45}/0.990$ | DIP-MCDO | $18.91/0.830$ | $\mathbf{24.76}/0.953$ | $\mathbf{27.72}/0.972$ | $\mathbf{29.09}/0.978$ |

Table 3 shows the test log-likelihood for all the methods and experimental settings under consideration. The peak signal-to-noise ratio (PSNR) and structural similarity (SSIM) index of posterior mean reconstructions are given in table 4. All methods show similar PSNR with the standard DIP (with TV regularisation) obtaining better PSNR in the very ill-posed setting (5 angles) and MCDO obtaining marginally better reconstruction in all others. Despite this, the linearised DIP provides significantly better uncertainty calibration, outperforming all baselines in terms of test log-likelihood in all settings. Figure 4 shows an exemplary character recovered from a simulated observation $y$ (using 20 angles and 5% noise) with both linearised DIP and DIP-MCDO along with their associated uncertainty maps and calibration plots. DIP-MCDO systematically underestimates

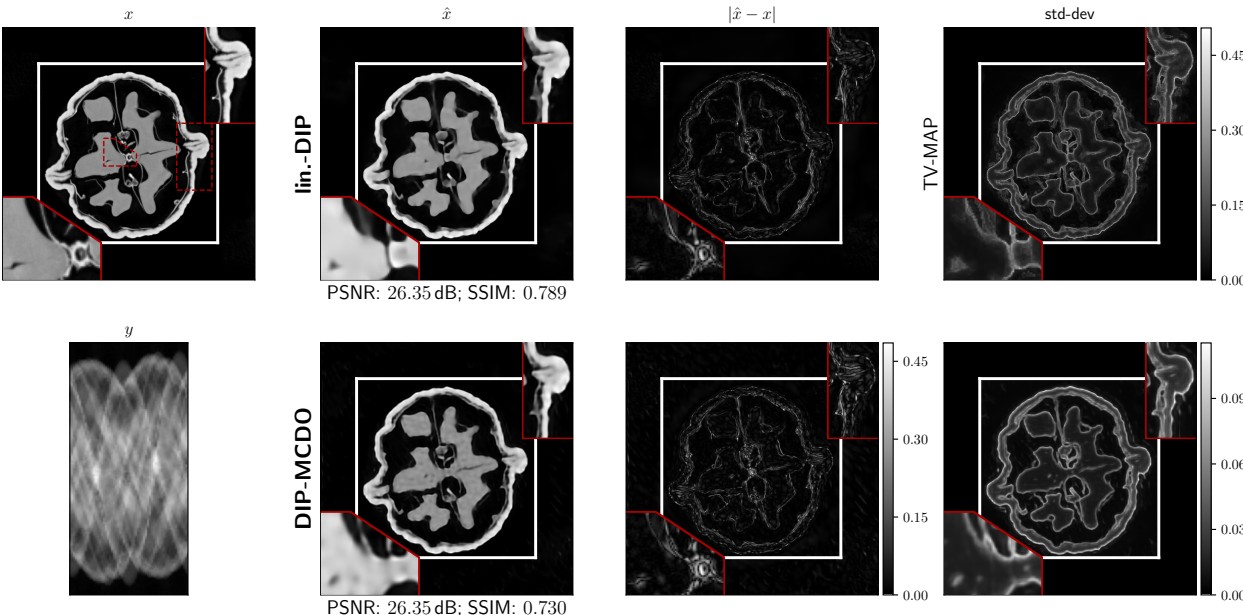

Figure 5: Reconstruction of a $501 \times 501 \, \mathrm{px}^2$ slice of a scanned Walnut using lin.-DIP and DIP-MCDO along with their respective uncertainty estimates. The zoomed regions (outlined in red) are given in top-left.

uncertainty for pixels on which the error is large, explaining its poor test log-likelihood. The pixel-wise standard deviation provided by linearised DIP (TV-MAP) better correlates with the reconstruction error.

### 7.1.3 Evaluating the fidelity of sample-based predictive covariance matrix estimation

We evaluate the accuracy of the sampling, conjugate gradient and low rank approximations to the predictive covariance $\Sigma_{x|y}$ discussed in section 6. We compute the exact predictive covariance with eq. (13) as a reference, which is tractable for KMNIST, and do not use patch-based approximations or stabilised covariance estimators. Table 5 shows that estimating $\Sigma_{x|y}$ using samples does not decrease the performance. Using a low-rank approximation to $J$ and computing linear solves with PCG lose at most 0.32 nats in test log-likelihood with respect to the exact one, but result in almost an order of magnitude speedup at prediction time.

Table 5: Evaluation of our approximate covariance estimation methods in terms of test log-likelihood over 10 KMNIST test images considering the 20 angle ($d_y = 820$) setting and using lin.-DIP (MLL).

| $\eta$ (%) | exact cov. eq. (13) | sampled cov. eq. (22) | sampled cov. ($\tilde{J}$) eq. (22) | sampled cov. ($\tilde{J}$ & PCG) eq. (22) |
|---|---|---|---|---|
| 5 | $2.80 \pm 0.06$ | $2.80 \pm 0.06$ | $2.68 \pm 0.09$ | $2.62 \pm 0.09$ |
| 10 | $2.26 \pm 0.06$ | $2.26 \pm 0.06$ | $2.21 \pm 0.06$ | $2.22 \pm 0.06$ |

Table 6: Test log-likelihood, PSNR and structural similarity (SSIM) on the Walnut. We compare all lin.-DIP variants with DIP-MCDO.

| | $1 \times 1$ | $2 \times 2$ | $10 \times 10$ | PSNR [dB] | SSIM |
|---|---|---|---|---|---|
| DIP-MCDO | 0.03 | 1.68 | 2.47 | 23.49 | 0.730 |
| lin.-DIP (MLL) | 2.09 | 2.25 | 2.43 | **26.35** | **0.789** |
| lin.-DIP (MLL, $\tilde{J}$ & PCG) | 1.88 | 2.05 | 2.24 | – | – |
| lin.-DIP (TV-MAP) | **2.21** | **2.40** | **2.60** | – | – |
| lin.-DIP (TV-MAP, $\tilde{J}$ & PCG) | **2.24** | **2.46** | **2.65** | – | – |

## 7.2 Linearised DIP for high-resolution CT

We now demonstrate the approach on real-measured cone-beam $\mu$CT data of a walnut (Der Sarkissian et al., 2019). We reconstruct a $501 \times 501 \, \mathrm{px}^2$ slice ($d_x = 251\,001$) using a sparse subset of measurements taken from 60 angles and 128 detector rows ($d_y = 7680$), using the U-Net in fig. 3 which has about 3 million parameters. Here, $\Sigma_{xx}$ is too large to store in memory and $\Sigma_{yy}$ too expensive to assemble repeatedly, and we use the full suite of approximations in section 6. Since the Walnut data is not quantised, jitter correction is not needed.

During MLL and Type-II MAP optimisation, many layers' prior variance goes to $\sigma_d^2 \approx 0$, cf. appendix D. This phenomenon is known as "automatic relevance determination" (Mackay, 1996; Tipping, 2001), and simplifies our linearised network, preventing uncertainty overestimation. We did not observe this effect when working with KMNIST images and smaller networks. We display the MLL and MAP optimisation profiles for the

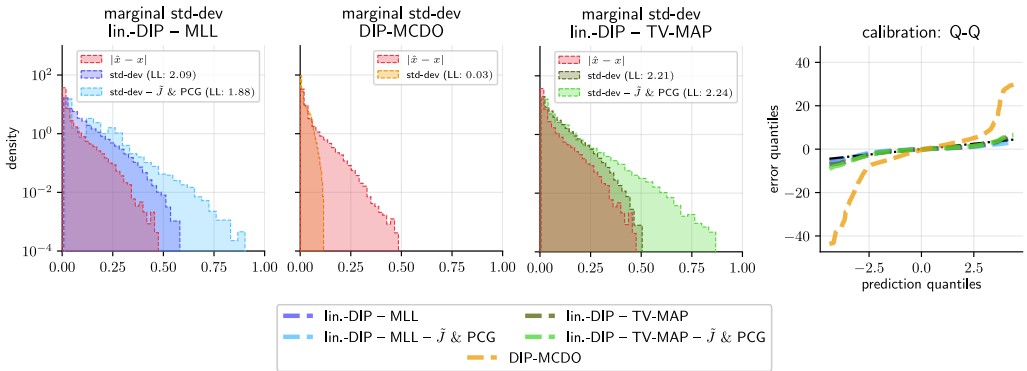

Figure 7: The comparison of uncertainty calibration: the pixel-wise error $|\hat{x}-x|$ overlaps with the uncertainties provided by the lin.-DIP. DIP-MCDO, instead, severely underestimates uncertainty. The scale of the pixel-wise standard deviation (std-dev) obtained including the TV-PredCP matches the absolute error more closely than when the hyperparameters are optimised without. Using $\tilde{J}$ & PCG results in overestimating uncertainty in the tails. LL stands for test log-likelihood.

active layers (i.e. layers with high $\sigma_d^2$) in fig. 6. Type-II MAP hyperparameters optimisation drives $\sigma^2$ to smaller values, compared to MLL. This restricts the linearised DIP prior, and thus the induced posterior, to functions that are smooth in a TV sense, leading to smaller error-bars, cf. fig. 7. As the optimisation of eq. (17) progresses, $\ell_1, \ell_{11}$ fall into basins of new minima corresponding to larger lengthscales. This results in more correlated dimensions in the prior, further simplifying the model.

Density estimation described in section 6.3, is conducted in double precision (64 bit floating point) since single precision led to numerical instability in the assembly of $\Sigma_{yy}$, and also in the estimation of off-diagonal covariance terms for larger patches. In table 6, we report test log-likelihood computed using a Gaussian predictive distribution with covariance patches of sizes $1 \times 1$, $2 \times 2$ and $10 \times 10$ pixels. Mean reconstruction metrics are also reported. Figure 5 displays reconstructed images, uncertainty maps and calibration plots. In this more challenging task, DIP-MCDO performs poorly relative to the standard DIP formulation eq. (5) in terms of PSNR. DIP-MCDO underestimates uncertainty, and its uncertainty map is blurred across large sections of the image, placing large uncertainty in well-reconstructed regions and vice-versa. In contrast, the uncertainty map provided by linearised DIP is fine-grained, concentrating on

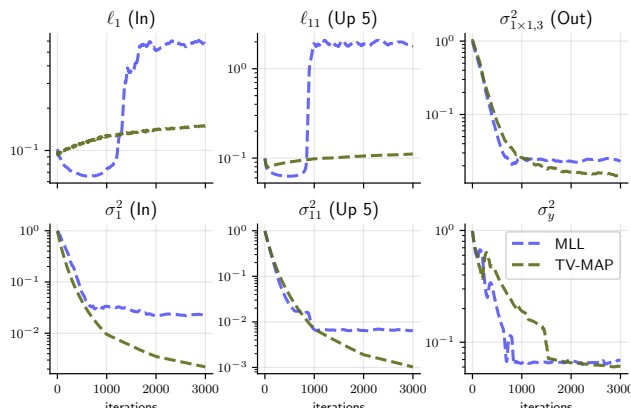

Figure 6: Optimisation trajectories for hyperparameters of the U-Net's first and last $3 \times 3$ convolutions $(\ell_1, \sigma_1^2, \ell_{11}, \sigma_{11}^2)$, last $1 \times 1$ convolution $(\sigma_{1\times1,3}^2)$ and noise variance $\sigma_y^2$ for the Walnut data.

regions of increased reconstruction error. Linearised DIP provides over 2.06 nats per pixel improvement in terms of test log-likelihood and more calibrated uncertainty estimates, as reflected in the Q-Q plot in fig. 7. Furthermore, the use of TV-PredCP prior for MAP optimisation yields a 0.12 nat per pixel improvement over the MLL approach. Interestingly, using low-rank Jacobians and PCG for sampling provides a small performance boost when using the TV-PredCP prior. Figure 7 reveals that these approximations result in uncertainty overestimation (a known issue (Antorán et al., 2023)) which is compensated by the more restrictive TV-PredCP prior.

## 8 Conclusion

We have proposed a probabilistic formulation of the deep image prior (DIP) that utilises a linearisation of the DIP network around the mode of the loss and a Gaussian-linear hierarchical prior on the network

parameters mimicking the total variation prior (constructed via the predictive complexity prior framework). The approach yields well-calibrated uncertainty estimates on tomographic reconstruction tasks based on simulated observations and real-measured $\mu$CT data. The empirical results suggest that both the DIP reparametrisation and the TV regulariser provide good inductive biases for high-quality reconstructions and well-calibrated uncertainty estimates. The method is shown to provide by far more calibrated uncertainty estimates than existing MC dropout approaches to uncertainty estimation with the DIP. However, this comes at a larger computational cost. Fortunately, since the first appearance of this work, Antoran et al. (2023) have developed techniques that reduce the cost of our linearised DIP inference by two orders of magnitude.

## Acknowledgements

The authors would like to thank Shreays Padhy, Marine Schimel, Alexander Terenin, Eric Nalisnick, Erik Daxberger and James Allingham for fruitful discussions. R.B. acknowledges support from the i4health PhD studentship (UK EPSRC EP/S021930/1), and from The Alan Turing Institute (UK EPSRC EP/N510129/1). J.L. was funded by the German Research Foundation (DFG; GRK 2224/1) and by the Federal Ministry of Education and Research via the DELETO project (BMBF, project number 05M20LBB). The work of BJ is partially supported by UK EPSRC grant EP/V026259/1 and a start-up fund from The Chinese University of Hong Kong. JMHL acknowledges support from a Turing AI Fellowship EP/V023756/1 and an EPSRC Prosperity Partnership EP/T005386/1. JA acknowledges support from Microsoft Research, through its PhD Scholarship Programme, and from the EPSRC. This work has been performed using resources provided by the Cambridge Tier-2 system operated by the University of Cambridge Research Computing Service (http://www.hpc.cam.ac.uk) funded by EPSRC Tier-2 capital grant EP/T022159/1.

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
