# OpenReview forum: "Uncertainty Estimation for Computed Tomography with a Linearised Deep Image Prior"
_TMLR — Accepted by TMLR_

### Review · Reviewer_hYVv · 2023-06-23

**Summary Of Contributions:**

Develops a technique to estimate the uncertainty associated with CT reconstructions formed by Deep Image Prior with TV regularization. The proposed technique works by linearizing (first order Taylor expansion) DIP around its optimized parameterized, which can be used to generate error bars for the reconstructions. The proposed method compares favorably to existing DIP uncertainty estimation techniques.


**Audience:**

Yes

**Claims And Evidence:**

Yes

**Requested Changes:**

If the authors could answer my question about p(x) vs p(x|\theta), I'm happy with the paper as is. While the paper's focus is relatively niche (uncertainty estimation for DIP), it makes a novel and well-validated contribution to that niche.


**Strengths And Weaknesses:**

Strengths

+Studies important problem

+Outperforms state-of-the-art DIP uncertainty estimation methods

+Generally well-written


Weaknesses

-Provides only an uncertainty estimate around a mode. If posterior is multimodal this won't be accurately reflected in the results. Paper convincingly argues this "shortcoming" improves interpretability as practitioners may not benefit from multiple modes anyway.

-Section 2 places a Gaussian prior on theta (and implicitly x), but this doesn't show in p(x) in equation (7). Is equation (7) more accurately described as p(x|\theta), with p(x)=p(x|\theta)*p(\theta)?

-Comparisons are restricted to DIP and data is toy-like (Walnut has simpler structure than typical CT scenes).



Typos:

Pg 4: unormalized-->unnormalized

Figures 4 and 5: One of the images seems to be missing a colormap and it's unclear if "TV-MAP" belongs.

---

> ### Author Response · Authors · 2023-08-11
> **Response to hYVv**
>
> We thank reviewer for their time in reading our paper, for their insightful comments and their suggestions. We greatly appreciate that the reviewer shares our enthusiasm for uncertainty estimation for Computed Tomography and that they are "happy with the paper as is".
>
> We go on to address the reviewer's questions and concerns.
>
> -----
>
> 1. $\mathbf{p(x)}$ **vs** $\mathbf{p(x|\theta)p(\theta)}$
>  The reviewer writes:
> "Section 2 places a Gaussian prior on theta (and implicitly $x$), but this doesn't show in $p(x)$ in equation (7). Is equation (7) more accurately described as $p(x|\theta)$, with $p(x)=p(x|\theta)p(\theta)$?"
>
> Thanks for pointing this out! Section 3, which contains equation 7, takes a step back and addresses non-DIP-based priors for image reconstruction. Thus, **the prior is placed directly on $x$ and not induced through a prior over DIP parameters $\theta$.**
>
> Section 3 aims to build intuition about how classical Bayesian image reconstruction priors can be approximated with Gaussians, which we will need when working with the linearised DIP. The reviewer is correct in general however; when working with the linearised DIP in the rest of the paper, the prior over x will be induced by a prior over the parameters $\theta$.
>
> **We clarify this at the beginning of section 3** with the language *"We commence by studying the construction of tractable non-DIP-based priors for CT reconstruction. The understanding gained will later allow us to incorporate TV-based priors into the linearised DIP framework."*
>
> ----
>
> 2. **Complexity of Walnut CT scan**
> We again quote the reviewer:
> *"data is toy-like (Walnut has simpler structure than typical CT scenes)"*
>
> This is an interesting comment that suggests some clarification in the main text.
>
> In academic research, CT reconstruction is most often studied in the medical context, likely due to the presence of existing collaborations with medical practitioners. However, **industrial micro-CT, i.e. the Walnut, is often more challenging from a technical standpoint.** We elaborate below:
>
>  -- **Realisticness:** Due to privacy constraints and use of proprietary scanning hardware, experiments using medical data almost always require that researchers simulate the scanning process post-hoc. Simulated forward operators and noise may be less challenging due to not capturing the nuance of real-data (e.g., complexity of operator geometry). The Walnut data we use consists of real measurements provided by Der Sarkissian et al. in ''A cone-beam X-ray computed tomography data collection designed for machine learning'' together with the full parameter set of their scanning hardware.
>
> -- **Complexity of the structure:** The Walnut data provides an arguably more challenging task than most medical CT due to the high-resolution of the micro-CT scan revealing the presence of small filaments and sharp discontinuities. This is not to say that *abdominal* CT does not present complex structure, which it does. However, we consider the Walnut to be on the more-challenging side of tasks, more so than limb or brain CT.
>
> ----
>
> 3. **Typos and other minor concerns:**
>
> * Thanks for pointing out typos; they are now fixed.
>
> * Regarding Figures 4 and 5, in both cases, **the colormap is shared between the empirical error $|\hat{x} - x|$ and the predictive standard deviation**. We have added a clarification in the figure's caption.
>
> * We will also clarify what TV-MAP refers to Type-II MAP optimisation of hyperparameters. We call it TV-MAP because the prior over the hyperparameters is TV-based.

---

### Review · Reviewer_955y · 2023-06-25

**Summary Of Contributions:**

This submission suggests an application of deep image prior (DIP) to be integrated with total variation regularisation for computed tomography. Authors add a post-processing step to the DIP-based CT image denoising in the framework of maximum a posteriori (MAP), with the initialised iteration point of the MAP optimisation problem as the output of a deep image prior and with other regularisation terms derived from total variation.

Technically, there is no novelty in the DIP design itself. The methodology innovation in this manuscript is the post-processing MAP step.

**Audience:**

Yes

**Broader Impact Concerns:**

No ethical implications or concerns noticed.

**Claims And Evidence:**

Yes

**Requested Changes:**

My specific questions on the manuscript:

(1) The DIP has a technicality that when to stop training is unapparent to decide. With such a complicated objective to optimise (equation 17 on page 7), deciding the early stopping seems a challenge. I do not see particular discussion made in the manuscript about the control of early stopping. It will be helpful for the audience to know authors' technique to manipulate the training setting.

(2) The computation of $\Sigma_{yy}^{-1}$ seems computationally insurmountable. It is unclear to me how authors manage to pull out the MAP optimisation process in the CT setting. The coordination between the updates made to parameters $\sigma$ and $\ell$ should be clearly explained.

(3) What are y-axis labels in figure 6?

(4) Why values are missing in table 6 for the latter three methods?

(5) Figure 7 seems to show that the DIP-MCDO has the deviation statistics regarding pixels more concentrated than linearised DIP methods. Is this not a desired property to have a reduced uncertainty?

(6) For sanity check, authors should clearly demonstrate the comparison between the results from "training the DIP to an optimal TV-regularised setting $\widehat{x}$" and that from he propose linearised DIP method.

Minor questions:

- What is "nats'?

- How is 'task-agnostic pretraining' set up in section 2.3?

- Define $J_d$ in equation 20.


**Strengths And Weaknesses:**

The benefit of such a formulation is the characterisation of uncertainty with respect to the denoised CT images, which can be of particular usefulness in the case of limited-angle measurement.

The disadvantage is that the formulation is highly computationally expensive and the practicality of the proposed methodology is put into doubt.

---

> ### Author Response · Authors · 2023-08-11
> **Response to 955y --- Main Concerns**
>
> Thank you for your review and relevant questions. We go on to address the reviewer's questions and concerns.
>
> ----
> **Weakness: Computational Cost**
> The reviewer writes: *"The disadvantage is that the formulation is highly computationally expensive and the practicality of the proposed methodology is put into doubt."*
>
> The reviewer is right that our hyperparameter optimisation is costly (35.4h as reported in Table 1.) We are transparent about this: the wall-clock time for every stage of our approach is reported in Table 1. We argue that **our method is complementary to cheaper but worse performing alternatives, like MC-dropout.**
>
> Additionally, since the appearance online of this paper, Antoran et. al. "Sampling-based inference for large linear models, with application to linearised Laplace" have introduced a novel SGD-based posterior sampling algorithm, which they apply to our Bayesian-DIP framework. **They reduce the cost of hyperparameter optimisation on the Walnut data to 12 minutes.**
>
>
> 1. **Stopping optimisation of Eq. 17.**
>
> The reviewer writes: *"The DIP has a technicality that when to stop training is unapparent to decide. With such a complicated objective to optimise (equation 17 on page 7), deciding the early stopping seems a challenge. I do not see particular discussion made in the manuscript about the control of early stopping. It will be helpful for the audience to know authors' technique to manipulate the training setting."*
>
> Thanks for this question! Unlike DIP training, **Type-II MAP optimisation does not require early stopping**. We want to truly find the maximum of this objective. Since we are optimising a relatively small number of prior hyperparameters (less than 100), we are not afraid of overfitting. We will clarify this explicitly in the main text.
>
> 2. **Computation of inverses: $\Sigma_{yy}^{-1}$**:
>
> We again quote the reviewer: *"The computation of  seems computationally insurmountable. It is unclear to me how authors manage to pull out the MAP optimisation process in the CT setting. The coordination between the updates made to parameters and should be clearly explained."*
>
> This is also a good question! The reviewer is correct that direct computation of $\Sigma_{yy}^{-1}$ would be insurmountable. Instead, as explained in 5.1, we always compute products of the shape $\Sigma_{yy}^{-1} v$, where $v$ is a vector. Solving $\Sigma_{yy}^{-1} v$ amounts to solving a system of linear equations which can be done **approximately (but to a very high accuracy)** with the **Preconditioned Conjugate Gradients method**. Conjugate Gradients only requires computing multiplications of the sort $\Sigma_{yy} v$ (notice the $\cdot^{-1}$ is not present). For this, we use the GPytorch Conjugate Gradients implementation (https://proceedings.neurips.cc/paper_files/paper/2018/file/27e8e17134dd7083b050476733207ea1-Paper.pdf).
>
> 3. **"What are y-axis labels in figure 6?"**
>
> Thank you for spotting this! The y-axis refers to the value of the hyperparameter, which is indicated in the title of each subplot.
>
> 4. **"Why values are missing in table 6 for the latter three methods?"**
>
> The reviewer is referring to the PSNR and SSIM values in the table.
>
> PSNR and SSIM only depend on the mean reconstruction and not on the uncertainty estimates. Our post-processing only adds uncertainty estimates. It does not change the mean reconstruction. As a result, **the values are the same for all linearised-DIP variants**. These metrics are only provided to show that the linearised-DIP is able to achieve better mean reconstructions than MCDO-DIP since linearised DIP is able to take advantage of modern improvements in DIP optimisation.
>
> 5. **"Figure 7 seems to show that the DIP-MCDO has the deviation statistics regarding pixels more concentrated than linearised DIP methods. Is this not a desired property to have a reduced uncertainty?"**
>
> This is a good question. A way to assess the quality of the estimated uncertainty is to check whether it is calibrated; that means whether the empirical errors in reconstruction match the uncertainty (in terms of standard deviation). In this case, **DIP-MCDO presents uncertainty much smaller than its reconstruction errors. It is underestimating the uncertainty.**
>
> 6. **"For sanity check, authors should clearly demonstrate the comparison between the results from "training the DIP to an optimal TV-regularised setting" and that from he propose linearised DIP method."**
>
> This is also a good question. Our post-processing does not change the reconstruction obtained by "training the DIP to an optimal TV-regularised setting". Our post-processing only adds uncertainty estimates. That means, we provide a standard deviation number for each pixel, which tries to predict how reliable the reconstruction of that pixel is. As a result, **there is no need to compare against the TV-regularised DIP reconstruction without our method, since they are both the same.**

---

> > ### Author Response · Authors · 2023-08-11
> > **Addressing minor concerns**
> >
> > **Additional questions:**
> >
> > * **What is "nats"?**
> >
> > Nats are "the natural unit of information". They are a base $e$ generalisation of bits. In particular, the differential entropy of a source (with associated probability density $p$) in nats is given by $-\int p(x) \log_{e} p(x)\,dx$.
> >
> > The test-log-likelihood represents how much information we would need to send to a receiver to communicate a test image to them, assuming that they have access to our probabilistic model, i.e. our Bayesian DIP. Differences between test log-likelihood values thus represent amount of information saved (not sent) and are thus measured in nats.
> >
> > See https://en.wikipedia.org/wiki/Nat_(unit) for nats and https://en.wikipedia.org/wiki/Differential_entropy for differential (the continuous generalisation of) entropy.
> >
> >
> > * **How is 'task-agnostic pretraining' set up in section 2.3?**
> >
> > Thanks for bringing this up! Since, all our DIP models use task-agnostic pre-training, we will provide a summary of this method in the appendix for completeness.
> >
> > Task-agnostic pretraining is an approach introduced by *Barbano et. al. 2021: An educated warm start for deep image prior-based micro CT reconstruction* by which the DIP is trained as a supervised model to reconstruct a dataset of synthetically generated images from their simulated CT scans. The dataset is generic, and thus the method is "task-specific".
> >
> > This approach initialises the DIP weights for standard DIP learning in a way that reduces the number of training steps required by over an order of magnitude.
> >
> > * **Define $J_d$ in equation 20.**
> >
> > Thanks for pointing this out! Not including this definition explicitly was an oversight which we now fixed.
> >
> > $J_d$ refers to the Jacobian of the DIP output with respect to the weights in layer $D$. It is defined analogously to $J$, that is $J_d = \frac{\partial x(\theta)}{\partial \theta_d}\_{\theta_{d}=\hat{\theta}_{d}}$.

---

### Review · Reviewer_M8N9 · 2023-09-12

**Summary Of Contributions:**

The authors present a novel method for uncertainty prediction in deep image prior (DIP) -based inverse imaging. Uncertainty estimates are provided by a Bayesian formulation of the inverse imaging problem combined with a linearized DIP model. The authors further improve the algorithm by proposing various approximations that greatly reduce computation time while incurring negligible performance reductions.

**Audience:**

Yes

**Broader Impact Concerns:**

While a broader impact statement is certainly welcome, I do not see any clear ethical concerns with the presented work that necessitates the addition of one.

**Claims And Evidence:**

Yes

**Requested Changes:**

*Inclusion of a related works section in the main work*: At present, the manuscript is missing a related works section. This is deferred to the appendix, presumably due to space restrictions. I find this solution somewhat unsatisfactory, since clearly delineating the novelty of the algorithm and placing it in the context of its surrounding literature is a crucial step to understanding a research paper. I do emphasize that this is a more minor concern -- perhaps the authors can at least mention that the related works are in the appendix?

*Discussion of limitations*: It may also be fruitful to describe the limitations of the present work. For example, how does the computational cost of the locally linear approximations related to other DIP-based inverse imaging algorithms? How restrictive is the Laplace approximation?

Minor:
"The discrepancy among plausible reconstructions in the posterior indicates uncertainty." <- This sentence is unclear to me. Are the authors just stating that $p(x|y)$ models the uncertainty of the reconstruction $x$ given the provided Bayesian model?

**Strengths And Weaknesses:**

**Strengths:**

- The proposed algorithm, as well as the presented computational enhancements, are simple and straightforward.
The model is adequately evaluated in a series of empirical experiments.
- According to the provided experiments, the calibration of the uncertainty estimates and the quality of the reconstructed images are both qualitatively and quantitatively competitive.

**Weaknesses:**

- Discussion of the limitations of the work. The authors note that the uncertainty estimate provided in this paper can only be obtained with respect to a locally linear approximation of the DIP model around its optimized parameters. Is this an adequate approximation? Moreover, the algorithm appears to be quite expensive in terms of wall-clock time (Table 1). Is this a unique weakness of the presented approach? How does it compare with competing methods?
- Limited empirical comparisons. Is there a reason why the paper only compares against a few of the papers discussed in the related works section (Appendix A)? Moreover, is it appropriate to leave the related works section in the Appendix?

---

> ### Author Response · Authors · 2023-09-17
> **Response to Reviewer M8N9 - Main Concerns**
>
> We thank the reviewer for their time reading our paper and for their comments and suggestions. We go on to address their questions and concerns.
>
> ---
>
> **Lack of related work in the main text:**
>
> We agree with the reviewer that **it would be better to have the related work in the main text**. The original decision to place it in the appendix was made to keep the paper to 12 pages. Given that this is not a strict requirement of TMLR, **we have moved the related work to section 2**, before the preliminaries. We have also included some new citations to relevant work that has been released since the submission of our paper.
>
> **Comparison with prior works:**
>
> In prior literature, **there are two probabilistic formulations of the DIP**. [Cheng et al., 2019] use Stochastic Gradient Langevin Dynamics, and [Laves et al., 2020] Monte Carlo (MC) dropout.
> However, in both cases, the primary focus is on preventing overfitting, a candent topic within the DIP literature, rather than accurately estimating uncertainty. Nevertheless, both methods can be used to obtain uncertainty estimates. In fact, dropout is the go-to uncertainty estimation method in the imaging community.
>
> **This is more clear in the updated draft, given that the related work is now included in the main text**.
>
> We compare against MC dropout in our experiments. **We also implemented SGLD but it exhibited very poor performance on both the KMNIST and Walnut datasets**. Section 6 reads "We attempted to include DIP-SGLD [Cheng et al., 2019] in our analysis, but were unable to get the method to produce competitive results on tomographic reconstruction problems."
>
> **Computational cost:**
>
> We quote the reviewer "Moreover, the algorithm appears to be quite expensive in terms of wall-clock time (Table 1). Is this a unique weakness of the presented approach? How does it compare with competing methods?"
>
> **The linearised Laplace method is more expensive than alternatives because posterior sampling and hyperparameter optimisation requires iteratively solving high dimensional linear systems. Alternative methods, like MC dropout, require multiple forward passes through the network, which is often much cheaper. However, the quality of the resulting uncertainty estimates tends to be very poor.**
>
> We have added the following sentence in the main text: *"While the proposed approach offers a well-calibrated alternative to uncertainty quantification within the DIP framework, it  introduces computational overhead when compared to poorly calibrated approximate schemes, such as MCDO, which only require a forward pass through the network to generate a single sample".*
> This statement highlights the computational difference between the proposed approach and the MCDO variant of the DIP.
>
> In response to the reviewer's suggestion, we have now included the following language in our conclusion: *"The methodology proposed in this work has been shown to provide more calibrated uncertainty estimates than existing MC dropout approaches. However, this comes at a larger computational cost. Fortunately, since first appearance of this work, [[Antoran et al., 2023]](https://arxiv.org/pdf/2210.04994.pdf) have developed techniques that reduce the cost of our linearised DIP inference by two orders of magnitude."*
>
> **Quality of approximation:**
>
> We again quote the reviewer: "The authors note that the uncertainty estimate provided in this paper can only be obtained with respect to a locally linear approximation of the DIP model around its optimized parameters. Is this an adequate approximation?" [...] "How restrictive is the Laplace approximation?"
>
> The reviewer is correct in pointing out that the linearised Laplace approximation can seem crude. Performing Bayesian inference with neural networks is extremely challenging, and until recently, there were no methods that performed well [[Ashukha et al., 2021]](https://arxiv.org/pdf/2002.06470.pdf). The linearised Laplace method is perhaps the first approach to consistently produce strong uncertainty estimation results [[Daxberger et al., 2021]](https://proceedings.neurips.cc/paper/2021/hash/a7c9585703d275249f30a088cebba0ad-Abstract.html).
>
> Although a fundamental analysis of the linearised Laplace method falls beyond the scope of our work, **we do highlight that unimodality may have the advantage of increased interpretability** [[Bhatt et al., 2021]](https://arxiv.org/pdf/2011.07586.pdf) with the sentence: "Even if we could obtain the full posterior, downstream stakeholders without expertise in probability are likely to have little use for it. A single reconstruction together with its pixel-wise uncertainty may be more interpretable to end-users."

---

> > ### Author Response · Authors · 2023-09-17
> > **Addressing a minor concern**
> >
> > **Clarification regarding posterior uncertainty:**
> >
> > Again quoting the reviewer: "The discrepancy among plausible reconstructions in the posterior indicates uncertainty''. This sentence is unclear to me. Are the authors just stating that $p(x|y)$ models the uncertainty of the reconstruction $x$ given the provided Bayesian model?"
> >
> > You are correct, $p(x|y)$ models the uncertainty of the reconstruction $x$. But how can we tell when one predictive distribution is more uncertain than another? One way is to *count how many reconstructions have similarly large probability under the posterior*. If only one reconstruction captures most of the probability mass, then we are quite certain that that is the right reconstruction. If no reconstruction has the most probability mass, but instead it is split evenly across multiple different reconstructions, then we are more uncertain. Of course this analogy assumes a discrete sample space, but the idea carries over to continuous sample spaces. In retrospect, perhaps the phrasing in our text was too loose. We will clarify by changing it to *"The presence of different reconstructions with high-probability under the posterior indicates uncertainty."*

---

### Author Response · Authors · 2023-09-17
**Summary of changes to paper**

We thank all reviewers for their time reading our work and their comments.

----

We have incorporated the feedback into an updated draft with the following changes:

* R M8N9: We have moved the related work from the appendix to the main text. It now constitutes section 2 of the paper. The related work contains an extended discussion of alternative methods for uncertainty estimation with the DIP.
* R M8N9 and 955y: We have added discussion on the computational cost of our approach to both section 5.4 and the conclusion.
* R M8N9: We have clarified the relationship between the Bayesian posterior and uncertainty in our preliminaries section.
* R 955y: We defined $J_d$ in equation 20.
* R hYVv: We make explicit that in section 3 (now 4) the prior is constructed directly over $x$ while in the rest of the paper, it is placed on $\theta$.
* R hYVv: Fixed typos you pointed out, added clarifications regarding the colormap of figures 4 and 5, and clarified that TV-MAP refers to Type 2 MAP with the TV prior.

---

### Decision · Action_Editor_oph3 · 2023-11-02

**Recommendation:** Accept with minor revision

**Comment:**

The paper is mostly good to go.
I only have the following requirement: Please add a formal description of the final algorithm proposed, taking into account the different steps. This can be in the main text or in the appendix.

**Audience:**

The work is sound, but conciderned as relatively niche, i.e. it applies to certain classes of inverse problems.

**Claims And Evidence:**

The proposed algorithm, as well as the presented computational enhancements, are clear and relatively well presented. The model is adequately evaluated in a series of empirical experiments.
According to the provided experiments, the calibration of the uncertainty estimates and the quality of the reconstructed images are  competitive.

---

> ### Author Response · Authors · 2023-11-30
> **Thanks.**
>
> We have now uploaded the camera-ready version with the requested algorithm included on page 9 of the main text.
>
> Best,
> Authors